# Bed-Exit Behavior Recognition for Real-Time Images within Limited Range

**DOI:** 10.3390/s22155495

**Published:** 2022-07-23

**Authors:** Cheng-Jian Lin, Ta-Sen Wei, Peng-Ta Liu, Bing-Hong Chen, Chi-Huang Shih

**Affiliations:** 1Department of Computer Science and Information Engineering, National Chin-Yi University of Technology, Taichung 411, Taiwan; cjlin@ncut.edu.tw (C.-J.L.); zejipa5218@gmail.com (B.-H.C.); 2College of Intelligence, National Taichung University of Science and Technology, Taichung 404, Taiwan; 3Fall Prevention Center, Department of Physical Medicine & Rehabilitation, Changhua Christian Hospital, Changhua 500, Taiwan; tasen@cch.org.tw (T.-S.W.); 105546@cch.org.tw (P.-T.L.)

**Keywords:** behavior recognition, images, bed exit

## Abstract

In the context of behavior recognition, the emerging bed-exit monitoring system demands a rapid deployment in the ward to support mobility and personalization. Mobility means the system can be installed and removed as required without construction; personalization indicates human body tracking is limited to the bed region so that only the target is monitored. To satisfy the above-mentioned requirements, the behavior recognition system aims to: (1) operate in a small-size device, typically an embedded system; (2) process a series of images with narrow fields of view (NFV) to detect bed-related behaviors. In general, wide-range images are preferred to obtain a good recognition performance for diverse behaviors, while NFV images are used with abrupt activities and therefore fit single-purpose applications. This paper develops an NFV-based behavior recognition system with low complexity to realize a bed-exit monitoring application on embedded systems. To achieve effectiveness and low complexity, a queueing-based behavior classification is proposed to keep memories of object tracking information and a specific behavior can be identified from continuous object movement. The experimental results show that the developed system can recognize three bed behaviors, namely off bed, on bed and return, for NFV images with accuracy rates of 95~100%.

## 1. Introduction

Human behavior recognition is a process consisting of object detection and tracking and has become the core component of intelligent applications such as security surveillance, bed-exit monitoring, and fall detection [1]. According to the techniques used to identify the target object, object detection approaches can be divided into two categories: sensor and image. In sensor-based techniques, the object can be detected by means of either fixed or mobile modules. Fixed modules, such as infrared (IR) sensors [2,3], hybrid IR and pressure sensors [4], and radio frequency identification (RFID) [5,6], detect the object in a limited range. On the other hand, mobile modules based on accelerometers and gyroscopes can be integrated into wearables, which are attached to the detected objects [7,8]. Image-based techniques are typically combined with deep learning model to detect a target object after a data training process. The common source of images can be video sequences [9,10], IR thermal images [11], and integrated images from RGB and IR cameras [12,13]. Compared with sensor-based techniques, image-based techniques can better preserve the spatial information for object detection and have the potential to achieve multi-object detection.

Based on deep learning techniques, three main stages are included in an image-based behavior recognition process, namely object detection, object tracking, and behavior recognition. Three stages can be associated with their respective deep learning techniques. In the object detection stage, a convolution-based model is generally used to extract features of the target object from an image. Popular object detection models include You Only Look Once (YOLO) [14] and Region-based Convolution Neural Network (R-CNN) [15,16]. Furthermore, the lightweight version of YOLO (e.g., Yolov3-tiny [17] or Yolov4-tiny [18]) is designed to realize real-time object detection in embedded systems and mobile devices. The object tracking stage utilizes the object detection results to monitor the object temporally and establishes a spatial track for the target object. The most well-known object tracking models are Simple Online and Real-time Tracking (SORT) [19] and its extended version, Deep SORT [20]. In SORT/Deep SORT, a Kalman Filter is adopted to predict and update the object track [21], while the Hungarian algorithm solves the matching problem between the predicted track and the current detected track in the presence of short-term object occlusion [22]. As the Kalman Filter aims to provide an object tracking solution based on a linearity assumption, particle filter solutions, such as Correlative Mean-Field (CMF) filter, can be used to track objects in a nonlinear system [23,24]. In addition to the typical CNN-based object detection and tracking models, the Plain Vision Transformer detector (ViTDet) uses a non-hierarchical backbone network for object detection purposes [25], and the Spike Transformer Network (STNet) performs single object tracking in an event-driven manner [26]. In the final stage, the spatial tracks recorded for target objects can correspond to a behavior, and a memory-based technique is required to recognize behaviors from the long-range dependencies of sequential data (i.e., spatial tracks). Typically, Recurrent Neural Network (RNN) [27] and Long Short-Term Memory (LSTM) [28] are useful models for predicting important events over time. Consequently, a behavior recognition system can be regarded as a combination of deep learning models with their respective data training, parameters, and computation.

This paper aims to design an image-based behavior recognition system for a bed-exit monitoring application. In the World Health Organization’s (WHO) “Global Report on Falls Prevention in Older Age”, almost 30~50% of people living in long-term care institutions fall every year, and 40% of them suffer from recurrent falls [29]. Other reports show that approximately one in three community-dwelling people aged over 64 falls every year [30,31,32]. Bed-exit monitoring can be the first line of defense in preventing the subsequent fall, injuries, and even death [33]. For caregivers, a well-designed bed-exit monitoring system is expected to reduce their additional burden, especially for patients with high fall risk [34]. In addition to the effectiveness of behavior recognition, efficiency is essential for a bed-exit monitoring system in the smart ward environment. Specifically, caregivers and medical institutions prefer a rapid system deployment without construction in the ward. The requirements of such rapid deployment include (1) installing, removing, and transferring the system as required; (2) staying connected with the smart ward system when installed; (3) a single-purpose application designed only for bed-exit monitoring to achieve fast personnel training and easy use in the institutions. These requirements imply that bed-exit behavior recognition should be done using a single device with mobility and wireless connectivity. The existing behavior recognition solutions rely on a complex system consisting of three stages. When the bed-exit monitoring device is required to be small and lightweight for rapid deployment, the complex system can impede real-time behavior recognition due to its complexity and overhead. Furthermore, data training for multiple stages can be a long process and this easily causes difficulties with follow-up system maintenance.

In this paper, a simplified behavior recognition solution designed for a low-cost embedded system is presented for bed-exit monitoring. As a branch of behavior recognition, bed-exit monitoring focuses on identifying bed-related behaviors, such as laying on the bed and leaving the bed, to prevent potential falls and the resultant injuries. To ensure efficiency, this simplified system considers the images captured by a narrow field of view (NFV). In NFV images, the object detection range is limited to a specific area (e.g., a bedside) or even to a part of the user’s body in such a way that only some specific behaviors (e.g., on bed and bed exit) are monitored. Compared with wide field of view (WFV), in which more spatial information can be obtained, NFV images have simpler scenes to facilitate the training and operation of the behavior recognition process. NFV images, however, are associated with more object detection failures and abrupt scene changes can incur a negative effect on effectiveness. To achieve a tradeoff between effectiveness and efficiency, the proposed system develops an NFV-based behavior recognition process with the following stages: object detection, status classification, and behavior recognition. In the object detection stage, a highly efficient model, namely Yolov4-tiny, is adopted to attain real-time object detection in the embedded systems. Next, the object position information is retrieved and different activity statuses can be classified in accordance with the current position track. The behavior recognition stage utilizes a status classification queueing (SCQ) system to keep the memories of input statuses among pre-defined behavior queues. As a result, the negative effects, such as discontinued object detection and abrupt position change, are eliminated, and behavior can be identified according to a series of continued activities. Performance is evaluated via experiments conducted in the ward environment and time complexity analysis. Furthermore, a system prototype is developed in the experiments to capture NFV images with four camera angles: horizontal high/low and vertical high/low. From the performance results, the proposed system can achieve a real-time bed-exit behavior recognition with low complexity, and the results associated with horizontal angles outperform those of vertical ones.

This paper is organized as follows. Section 2 describes the concept of behavior recognition using NFV images, the proposed simplified behavior recognition system in embedded systems, and its bed-exit monitoring application. The experimental and analytical results are reported in Section 3. Section 4 gives the conclusions and future works.

## 2. Materials and Methods

### 2.1. Images with a Narrow Field of View

Figure 1 illustrates different object tracking ranges to monitor bed-related behaviors in the ward environment. A wide field of view tends to obtain more activity information by covering as wide an area as possible. Consequently, WFV images can contribute to the effectiveness of behavior recognition and are beneficial to tracking various behaviors such as bed exit and falling. IBy contrast, a narrow field of view aims to cover an area of interest where only the target is active and can obtain a simple scene for behavior recognition purposes.

Figure 2 shows the NFV system prototype used in this paper. The main components consist of a primary device, a secondary device, and a mobile stand. The primary device is in charge of user input (buttons and microphone), user output (display and speaker), and communication (Wi-Fi and call bell socket). On the other hand, the secondary device is connected to a camera and receives images to conduct the behavior recognition process. The recognition results are then sent to the primary device for the potential follow-up notification to the institution. Both primary and secondary devices are attached to the mobile stand. Accordingly, this prototype provides a post-installation feature and can be deployed or removed as required. Furthermore, the camera device can be adjusted as shown in Figure 2b,c to capture NFV images from various angles. Figure 3 presents images captured from four different angles: horizontal high, horizontal low, vertical high, and vertical low. Referring to Figure 1, the horizontal angles cover the bed region, while only a portion of the body is covered in the images captured by vertical angles. From Figure 3, it can be seen that NFV images with limited spatial information can impede object identification and tracking, and user activities become unpredictable.

### 2.2. Behavior Recognition for NFV Images

In this paper, a three-stage behavior recognition process is developed for NFV images: object detection stage, status classification stage, and behavior recognition stage. The object detection stage aims to identify the application-specific objects from a series of images. In order to adapt to the narrow field of view, it is beneficial to detect multiple objects in accordance with the scenarios defined in the application. In bed-exit monitoring applications, for instance, the detected objects can be “head” and “body” of a target user. When the object is successfully detected, its coordinate in the image is generally labeled to indicate the position of the detected object. Based on the object detection results, the status classification stage can further track the object in accordance with its position changes, and an activity status is determined. Both object detection stage and status classification stage are performed to generate their respective outputs per image. The behavior recognition stage continuously monitors the status output to identify a user behavior within a specified duration.

Figure 4 presents a status classification queueing (SCQ) system proposed in this study to realize the behavior recognition stage for NFV images. The proposed SCQ system consists of four components: input queue, classifier, output queue, and controller. The input queue and output queue are data buffers with a queue structure. That is, the buffered data are processed in a first-in-first-out manner. The data unit is the activity status from the status classification stage. Each status result enters the input queue and is then assigned by the classifier to one among *N* output queues according to its status. That is, the status *i* is in the output queue *i* with a length *L_i_*. Meanwhile, the controller drops one status from the head of its output queue to manage the number of buffered statuses in SCQ. The number of statuses, *M*, can be regarded as a status window size specified by a time slot of *T* seconds:(1)$$M=R\times T,\;\mathrm{subject}\;\mathrm{to}\;M\leq \sum_{i=1}^{N}L_{i}\;\mathrm{and}\;M\geq \max\left\{L_{i}\right\},$$
where *R* is the frame rate estimated in frames per second (fps). The queue length determines the recognition sensitivity for the behavior associated with it. Generally, a shorter queue length corresponds to a faster activity or a critical behavior requiring early notification. The controller continuously monitors *N* output queues. When one of them is full, the SCQ system can determine a user behavior associated with the full queue. Immediately after the behavior is identified, the controller empties all output queues to start a new recognition process. The next sub-section presents a bed-exit monitoring application based on this simplified behavior recognition process, including the proposed SCQ, to identify bed-related behaviors from NFV images.

### 2.3. Bed-Exit Application

This section describes a bed-exit application based on a three-stage behavior recognition system, for NFV images captured by the prototype shown in Figure 2. In the object detection stage, Yolov4-tiny is adopted to identify the head and trunk of a monitored user. As the object is detected in an NFV image, a bounding box with the position information of X and Y coordinates can be available to the status classification stage. Figure 5 shows the object detection results for NFV images captured from the horizontal low angle. Figure 5a–d indicates a series of bed-related activities including laying on the bed, turning over, getting up, and walking away from the bed. In Figure 5a–c, the detected objects are the head labeled in the pink bounding boxes, while Figure 5d detects the human trunk labeled in the green box.

Denote the X and Y coordinates centered in the bounding box as x_center and y_center, respectively. The center position of the bounding box is
(2)P={x_center , y_center}.

Given a time slot, *W*, in seconds, the number of images within *W* for an image sequence with *R* frames per second can be computed as
(3)uF=W×R.

Then the center position vector for bounding boxes of *uF* images is expressed as
(4)$$FP=\left\{P_{1},P_{2},...\;,P_{uF-1},P_{uF}\right\}.$$

In order to observe the *FP* changes, Figure 6 presents a real-time trace of the image sequence captured from the horizontal low angle. By inspecting the image sequence, we can differentiate user behaviors into four categories: laying on the bed, exiting the bed, nobody, and back to bed. These four behaviors are repeated twice and finally end with the “laying on the bed” behavior. Accordingly, a total of 9 behaviors are numbered from 1 to 9 in Figure 6. In the case of “nobody” behavior (numbered as 3 and 7 in Figure 6), the bounding box is absent and therefore no information about X and Y coordinates is available. In the behavior case labeled as 1 and 2 (i.e., laying on the bed and exiting the bed), however, three events labeled as A, B, and C are observed to have the same situation as “nobody” behavior. This is because the head and the trunk can be missed or be incomplete using a narrow field of view. Figure 7 shows the object tracking results for another image sequence captured from the vertical low angle. Following similar observations as in Figure 6, two object missing events occur in Figure 7: event A in the “back to bed” behavior numbered as 4, and events E~G in the “laying on the bed” behavior numbered as 9. Additionally, abrupt position changes are common for NFV images. In Figure 6, the “laying on the bed” behavior, numbered as 1, has frequent position level fluctuations, especially for the X coordinate. In Figure 7, position changes become even more abrupt. From Figure 7, it can be seen that (1) for the behaviors numbered as 4, 5, and 6 (i.e., “back to bed”, “laying on the bed”, and “exiting the bed”), three behaviors are performed in order within a short duration; (2) events B~D exhibit fast XY curve fluctuation indicating that the detected object appears and disappears suddenly; (3) the “laying on the bed” behavior numbered as 9, has position level fluctuations. This implies that diverse user activities such as turning over and getting up exist when the target user is on the bed.

In the status classification stage, a status, $\hat{S}$, can be defined in accordance with the distance between two positions as follows:(5)$$\hat{S}=\{\begin{array}{c} 1,\;\;if\;\;\bar{P_{i}P_{0}}-\bar{P_{i-1}P_{0}}\geq \varepsilon ,\;\;\\ 0,\;\;if\;\;\left|\bar{P_{i}P_{0}}-\bar{P_{i-1}P_{0}}\right|<\varepsilon ,\\ -1,\;\;if\;\bar{P_{i-1}P_{0}}-\bar{P_{i}P_{0}}\geq \varepsilon ,\;\;\\ -2,\;\;No\;bounding\;box,\;\; \end{array}$$
where *ε* is the distance threshold and *P*_0_ represents an initial object position. It is noted that the origin of an image (i.e., (0,0)) is located at the top left point/pixel. Referring to Figure 5a, the user laying on the bed initially has a smaller value of the X coordinate and a larger value of the Y coordinate. During the period of exiting the bed, the value of the X coordinate is gradually increased, and the value of the Y coordinate becomes smaller. The status of $\hat{S}$ = 1 indicates that the target user moves away from the bed since the distance between the current position and the initial position is larger than the distance between the previous position and the initial position by *ε*. On the contrary, the status of $\hat{S}$ = 1 means that the user moves back to the bed when the distance between the current position and the initial position decreases from image *i*−1 to *i*. When the distance between two neighboring positions is smaller than *ε*, the intention for a user to move toward/back the bed stays neutral and the status $\hat{S}$ is set to 0. Finally, the status of $\hat{S}$ = −2 indicates that the bounding box has been missed. In Equation (5), the initial position (*P*_0_) can be either a predefined value when the user remains stationary on the bed, or just set to the origin.

From the above-mentioned discussion, the status classified by a short-term position change may not correspond to the user behavior in the presence of abrupt position changes. In the behavior recognition stage, SCQ is utilized to differentiate the status associated with current behavior from that induced by abrupt position changes. In this study, four statuses defined in Equation (5) are injected into queues {*Q_j_*}, where *j* = 1~4. For Q_1_~Q_4_, the behaviors associated with their corresponding queues are “off bed”, “on bed”, “return”, and “nobody”, respectively. According to the queueing discipline of SCQ, a behavior is recognized when buffered statuses in their corresponding queue are full. Let the queue length of *Q_j_* be *L_j_* and given the number of statuses *M*, the behavior recognition of SCQ can be expressed as
(6)$$\hat{E}=\{\begin{array}{c} off\;bed,\;\;if\;\overset{i-M+1}{\underset{k=i}{\sum }}\hat{S_{k}}=L_{1},\;\;\;\;\;\;\;\;\;\;\;\\ on\;bed,\;\;if\;\;\overset{i-M+1}{\underset{k=i}{\sum }}\left(1-\hat{S_{k}}\right)=L_{2},\;\;\;\;\;\\ return,\;\;if\;\overset{i-M+1}{\underset{k=i}{\sum }}-\hat{S_{k}}=L_{3},\;\;\;\;\;\;\;\;\;\\ nobody,\;if\;\overset{i-M+1}{\underset{k=i}{\sum }}-\left(\hat{S_{k}}/2\right)=L_{4}.\;\;\;\;\;\; \end{array}$$

Based on (6), the statuses associated with a specified behavior can be accumulated to obtain an accurate observation of recent user activities. Meanwhile, the statuses representing abrupt position changes caused by a missed bounding box or object detection failure are omitted by means of putting them into other queues. All queues are emptied to restart another behavior recognition process after a behavior is determined.

## 3. Results

### 3.1. Experiment

In this study, a series of experiments were conducted in the ward environment. The system prototype presented in Figure 2 was deployed at the head of the bed, and four different camera angles were adopted to capture images with a frame rate of 15 frames per second and a resolution of 640 × 360. The input images were then adjusted to 416 × 416 for object detection by Yolov4-tiny. Two participants, each of whom wore clothes in five different colors, were involved in the experiments to establish a data set of 10 video sequences for every camera angle. Consequently, a total of 40 sequences were available for data training and performance evaluation purposes. The data set had 1213, 1319, 1124, and 861 images for horizontal high, horizontal low, vertical high and vertical low angle, respectively. The image ratio used for data training and evaluation is 8:2. In our system prototype, an Nvidia Jetson Xavier NX with 384 CUDA core and 8G memory operates as the secondary device to realize the three-stage behavior recognition process presented in Section 2.2. In order to evaluate the timely recognition of NFV images, lengths of all queues and the number of buffered statuses (*M*) are fixed to 15 (images/statuses) in the SCQ. That means the minimum response time to recognize a behavior will be one second.

For NFV images, multi-object detection facilitates the subsequent object tracking and behavior recognition stages. As mentioned in Section 2.3, the main detected objects in this study are the head and human trunk. Typically, the detection results can be divided into four classes: true positive (*TP*), false positive (*FP*), true negative (*TN*), and false negative (*FN*). Based on these detection classes, the detection accuracy (A) can be computed
(7)$$A\;=\frac{TP+TN}{TP+TN+FP+FN}.$$

Table 1 presents the basic object detection results for different camera angles. In additional to the four NFV angles, the “diagonal high” angle is further considered as a WFV case for performance comparison purposes. Figure 8 shows the snapshots from a “diagonal high” video sequence. The data set of diagonal high angle has 2460 images and the ratio of data training to evaluation is 8 to 2. For horizontal angles, both the high and low angles can attain an accuracy rate of 85% and even higher. In the case of horizontal high angle, the successful detection rate for the human trunk is higher than that of the horizontal low angle because the higher angle can better capture the trunk as the user walks around the bed. For vertical angles, the accuracy rate of successfully detecting the head can be higher than 95%. The vertical low angle, however, fails to detect the human trunk as the user gets up from the bed. Compared with NFV images captured from horizontal and vertical angles, the diagonal high angle covers an area as wide as possible to obtain better object detection performance. Specifically, accuracy rates of 99% and 96% are attained for head detection and human trunk detection, respectively.

Table 2 shows the more detailed results of the object detection stage. For each class among TP, FP, and FN, the detection results of “head” and “trunk” are counted together to observe the object detection performance of five different camera angles. The detection precision (P), Recall (R), and F1-score (F1) are given in order by:(8)$$P\;=\frac{TP}{TP+FP}.$$
(9)$$R\;=\frac{TP}{TP+FN}.$$
(10)$$F1=\frac{2\times P\times R}{P+R}.$$

From Table 2, the vertical angles have higher values than horizontal angles in terms of *P*, *R,* and *F1-score*. This is based on a fact shown in Table 1, that the vertical angles can better detect the head as the user lays on the bed. Except for the vertical low angle, two horizontal angles (i.e., high and low) and a vertical high angle achieve nearly the same mean average precision (*mAP*) value. The vertical low angle has a lower *mAP* value of 49% due to the poor detection of the human trunk. Although the three angles with high *mAP* values are able to support object detection for NFV images, the diagonal high angle achieves the highest values among all performance metrics {*P*, *R*, *F1*, *mAP*}.

As discussed above, NFV images can impede successful object detection and are expected to become an obstacle to behavior recognition accordingly. Obviously, the object detection failure would spread to the status classification stage where the object is further tracked and classified into different activity statuses. Consequently, the status track becomes more abrupt or discontinuous. The proposed SCQ aims to keep the memories of statuses based on a multi-queue architecture and predicts the user behavior labeled by a full queue. In the bed-exit monitoring application, we focus on three main user behaviors: on bed, off bed, and return to the bed. For each video sequence in our data set, the target user first lays on the bed, then walks away from the bed, and finally returns to the bed. This scenario is repeated once. Figure 9 shows the behavior recognition results for the video sequence presented in Figure 6. In order to ensure a good presentation quality, only the X-axis trace is plotted, and the three bed-related behaviors are expressed using behavior indexes. “on bed”, “off bed”, and “return” behaviors are numbered in order as 0, 1, and −1. The “nobody” behavior is indexed as 2. In the behavior case numbered as 1 (i.e., laying on the bed), SCQ gives a behavior index of 0 in the presence of object detection failure events A and B. In the end of behavior case numbered as 2 (i.e., exiting the bed), SCQ can successfully recognize “off bed” behavior with index = 1, while identifying “return” behavior (index = −1) in the beginning period of the behavior case numbered as 4 (i.e., back to bed). Similar results can be observed in behavior cases 6 and 8. In order to evaluate the corresponding performance for three bed behaviors, each video sequence is further divided into four “on-bed” clips, two “off-bed” clips, and two “return” clips. In the experiment, the amount of “on-bed” clips used to evaluate performance is 10 × 4 = 40. On the other hand, both the total amount of “off-bed” clips and “return” clips are the same as 10 × 2 = 20. The final behavior recognition results are presented in Table 3. From Table 3, the proposed SCQ can identify “on-bed” and “off-bed” behaviors with a successful recognition rate of 100% for all horizontal and vertical angles. In identifying “return” behavior, SCQ can achieve a successful recognition rate of 100% for horizontal angles, while obtaining a successful recognition rate of 95% for vertical angles. The images captured by vertical angles have a narrower field of view and fail to detect the object (head or trunk) for SCQ during a “return” behavior. Based on a similar deployment scenario, where the monitor device is installed at the head of the bed, the sensor-based work presented in [34] adapts an infrared array to differentiate the “off bed” behavior from body activities on the bed. According to the results reported in [34], an accuracy rate of about 99% is attained in recognizing “off bed” behavior. In addition to the fact that the proposed SCQ achieves an accuracy rate of 100% for “off bed” behavior, SCQ can recognize more bed-related behaviors, such as “on bed” and “return”, thanks to the better spatial information obtained by images. To conclude, SCQ is effective in identifying bed-related behaviors from NFV images and the horizontal angles are the preferred configuration in deploying an NFV-based bed-exit monitoring system.

### 3.2. Complexity Analysis

In this section, the time complexity of the proposed behavior recognition system is calculated and compared with that of the general one. For general behavior recognition systems, the object detection stage adopts a CNN-based method to find the features of the target object. The complexity of CNN is mainly caused by the convolution operation, the number of kernels, and the memory access [35]. The number of convolution operations per input grows quadratically with an increased kernel size. Accordingly, the complexity of *d* convolution layers can be estimated as
(11)$$O\left(\left(\sum_{i=1}^{d}C_{i-1}\times S_{i}^{2}\times F_{i}\times M_{i}^{2}\right)\times l\times e\right),$$
where *C_i−_*_1_ is the number of input channels of layer *i*, *S_i_* is the spatial size of the filter, *F_i_* is the number of filters of layer *i*, *M_i_* is the spatial size of the output feature map for layer *i*, *l* is the input length, and *e* is the number of epochs. In the object tracking stage, some general systems, such as SORT or Deep SORT, use the advanced technique, to facilitate behavior recognition. In Deep SORT, the Kalman Filter and Hungarian Algorithm are two primary components used to obtain the spatial track from the object detection results. According to the work reported in [36], the complexity of the Kalman Filter in an iteration is given by
(12)$$O\left(9\times H^{2.367}+10\times H^{2}+5\times H\right),$$
where *H* is the state size in the Kalman Filter. For the Hungarian Algorithm, its complexity was proved to be strongly polynomial, O(n^4^) [37], while Edmonds and Karp presented a faster version of the algorithm to achieve a running time of O(n^3^) [38].

In the behavior recognition stage, LSTM is considered a popular method of identifying user behavior based on sequential spatial data. The traditional LSTM contains memory blocks in the recurrent hidden layer, and each block is controlled by memory cells, gates, and peephole connections. Among LSTM gates, an input gate controls the flow of input into the memory cell, an output gate controls the present flow of output (i.e., weight) into the LSTM network, and a forget gate can reset the cell state. The peephole connection records the timing of output by linking gates in the same block. The idea behind LSTM is to allow temporal information to persist or be forgotten in its internal state. Generally, LSTM can be trained using the Back Propagation Through Time (BPTT) technique. For each time step, LSTM requires O(1) per weight to obtain its computational results [28,39]. Denote *w* as the number of weights and the total complexity of LSTM becomes:(13)O(w×l×e).

In the proposed three-stage behavior recognition process, the object detection stage employs CNN-based Yolov4-tiny and obtains the same time complexity as (10). As shown in Equation (5), the status classification stage calculates the distance between two positions twice, and its complexity can be O(1) based on the fact that the input size of the distance formula is fixed to two in each time step. Note that the advanced object tracking technique in general behavior recognition systems also involves distance computation, and the corresponding complexity can be ignored when compared to the large amount of computation contributed by the Kalman Filter and Hungarian Algorithm. Finally, the proposed SCQ performs an enqueue/dequeue operation or empties the queue in each time step, and its complexity is O(1) accordingly. From the statistics from the experiment results, the three-stage behavior recognition process conducted for bed-exit monitoring can operate in NX at an average frame rate of 34.65 frames per second. An average value of 35.88 fps is attained when only the object detection stage operates. Consequently, the performance gap caused by the complexity of the status classification stage and the behavior recognition stage is 1.23 fps on average.

## 4. Discussion

This paper aims to establish an image-based behavior recognition system in a real-world ward to identify bed-related behaviors when dealing with practical issues such as fast deployment and system maintenance. This means that for a typical behavior recognition process, its data training and computational cost need to be limited so as to realize the bed-exit system on an embedded system. In order to achieve efficiency, images are captured with a narrow field of view to obtain a simple scene for behavior recognition. Furthermore, a simplified three-stage behavior recognition process is presented to ensure recognition effectiveness: (1) in stage 1, multi-object detection based on Yolov4-tiny is conducted to enhance the detection performance when using a narrow field of view; (2) in stage 2, the changes in object position are classified into different statuses to achieve simple object tracking; and (3) a queue-based behavior recognition approach, namely SCQ, is proposed in stage 3 to deal with the problems associated with a narrow field of view, such as discontinuous and abrupt position changes, by means of keeping the memories of object tracking statuses.

The experiment results show that the object detection of wide-area images outperforms that of images with a narrow field of view, and the proposed SCQ can efficiently recognize bed-related behaviors from NFV images. For images captured from horizontal angles, the successful recognition rate of 100% can be attained in terms of “on bed”, “off bed”, and “return” behavior. On the other hand, the successful recognition rate associated with the vertical angles is 100% for “on bed” and “off bed” behavior, and 95% for “return” behavior. In addition, te complexity analysis indicates that stages 2 and 3 in the proposed behavior recognition process are conducted with very low complexity, and therefore, the resultant bed-exit monitoring system can achieve real-time processing at nearly 35 fps on average using an Nvidia NX testbed.

## Figures and Tables

**Figure 1 sensors-22-05495-f001:**
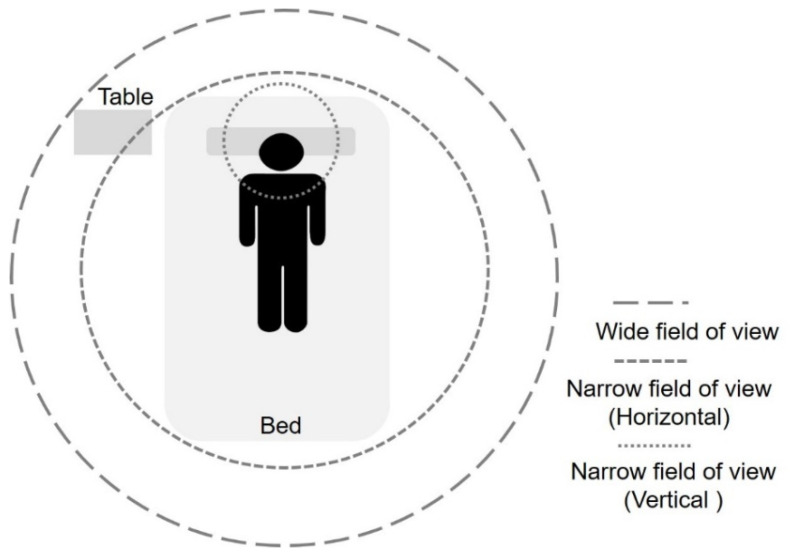
Object (human body) tracking range.

**Figure 2 sensors-22-05495-f002:**
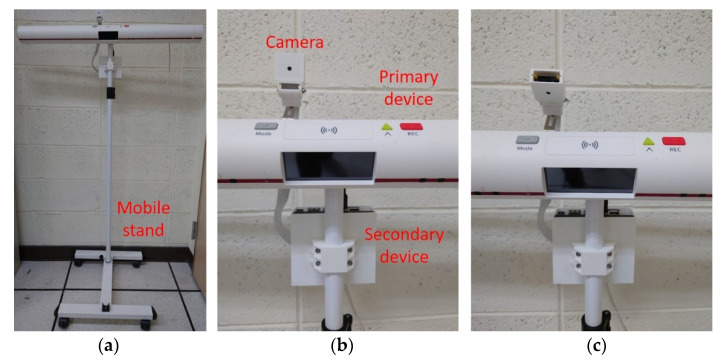
System prototype: (**a**) system with mobile stand; (**b**) camera capturing a horizontal view; (**c**) camera capturing a vertical view.

**Figure 3 sensors-22-05495-f003:**
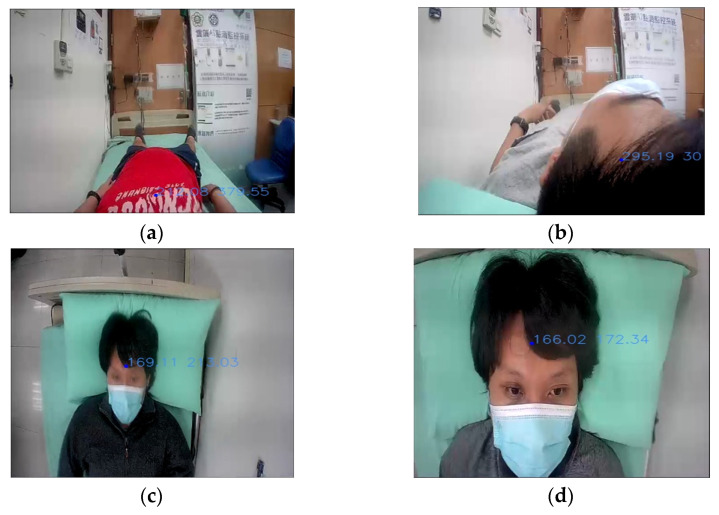
Images with four angles: (**a**) horizontal high; (**b**) horizontal low; (**c**) vertical high; (**d**) vertical low.

**Figure 4 sensors-22-05495-f004:**
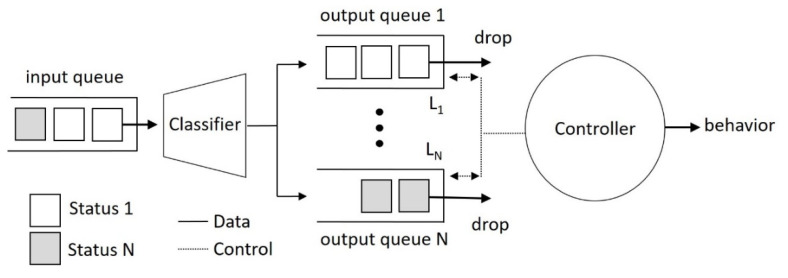
Status classification queueing system.

**Figure 5 sensors-22-05495-f005:**
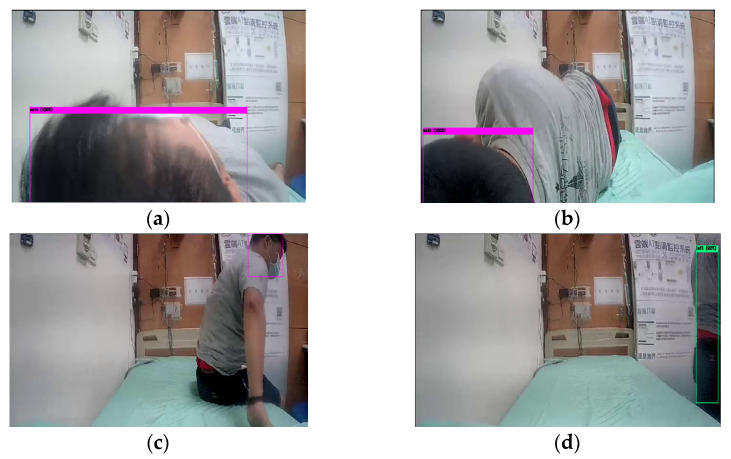
Object detection with bounding box in pink (**a**–**c**) and green (**d**).

**Figure 6 sensors-22-05495-f006:**
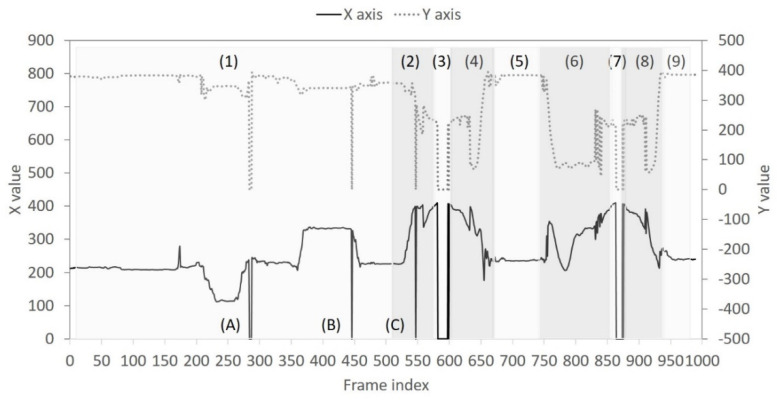
Real−time trace of XY coordinates for the bounding box from the horizontal low angle: numbers (1)~(9) stand for different statuses; letters (A)~(C) represent detection failure events.

**Figure 7 sensors-22-05495-f007:**
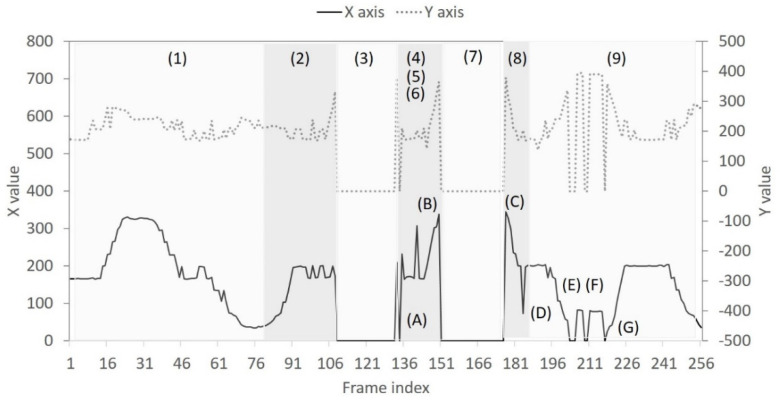
Real−time trace of XY coordinates for the bounding box from the vertical low angle: numbers (1)~(9) stand for different statuses; letters (A) and (E)~(G) represent detection failure events; letters (B)~(D) represent scene change events.

**Figure 8 sensors-22-05495-f008:**
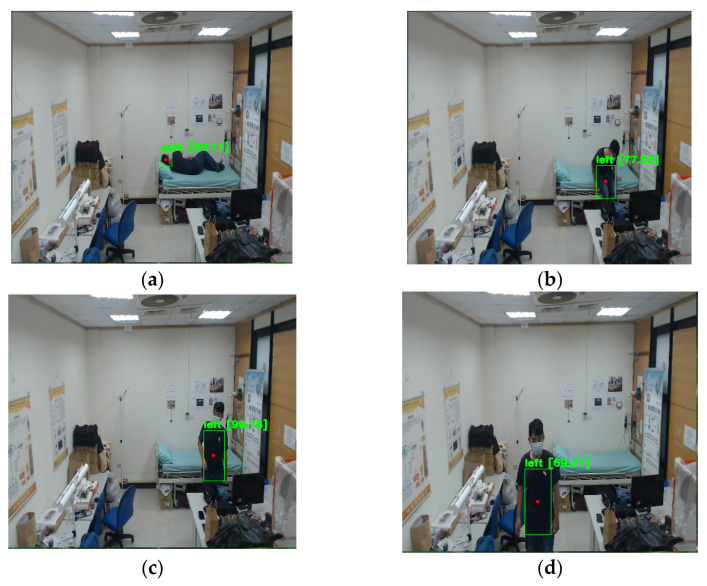
Images from the diagonal high angle: (**a**) lay on the bed; (**b**) get up and turn around; (**c**) sit on the edge of bed; (**d**) exit the bed.

**Figure 9 sensors-22-05495-f009:**
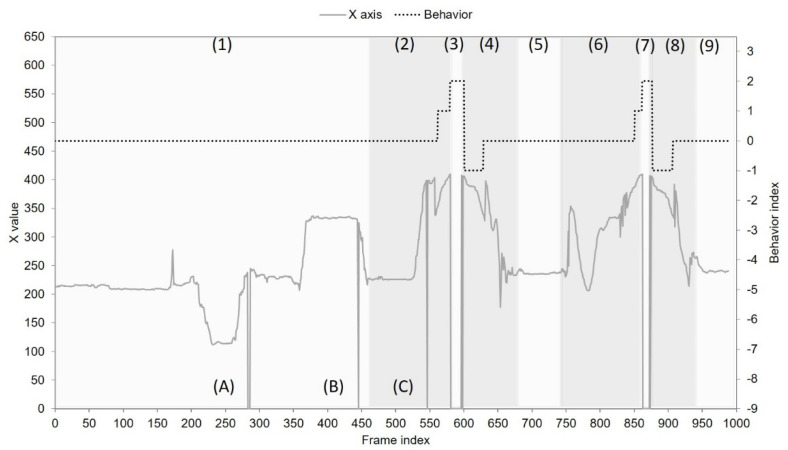
Behavior index trace from the horizontal low angle: numbers (1)–(9) stand for different statuses; letters (A)–(C) represent detection failure events.

**Table 1 sensors-22-05495-t001:** Head and trunk detection results for different camera angles.

Camera Angle	Object	TP	FP	A
Horizontal High	Head	159	46	86%
	Trunk	103	5	93%
Horizontal Low	Head	149	48	85%
	Trunk	112	25	85%
Vertical High	Head	218	15	96%
	Trunk	67	20	85%
Vertical Low	Head	154	1	95%
	Trunk	x	x	x
Diagonal High	Head	1817	2	99%
	Trunk	763	39	96%

**Table 2 sensors-22-05495-t002:** Object detection statistical results for different camera angles.

Camera Angle	P	R	F1	TP	FP	FN	mAP
Horizontal High	84%	85%	85%	262	51	45	90%
Horizontal Low	84%	85%	82%	261	73	45	88%
Vertical High	89%	92%	90%	285	37	24	90%
Vertical Low	99%	90%	95%	154	1	17	49%
Diagonal High	98%	98%	98%	2580	41	51	98%

**Table 3 sensors-22-05495-t003:** Behavior recognition statistical results for different NFV camera angles.

Camera Angle	On Bed	Off Bed	Return
Horizontal High	100% (40/40)	100% (20/20)	100% (20/20)
Horizontal Low	100% (40/40)	100% (20/20)	100% (20/20)
Vertical High	100% (40/40)	100% (20/20)	95% (19/20)
Vertical Low	100% (40/40)	100% (20/20)	95% (19/20)

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
