# Peer review of "Bed-Exit Behavior Recognition for Real-Time Images within Limited Range"

_sensors, 2022, doi:10.3390/s22155495_

Round 1
Reviewer 1 Report
The paper proposed bed-exit monitoring system based on 3 cascade modules: object detection, status classification, and behavior recognition. The target application of the proposed system is to be used for monitoring patients with high fall risk. The proposed application is very interesting and beneficial. However, there are some concerns that shall be addressed to strengthen the paper. These are the following.
In figure 5, input queue and output queue shall be defined.
In lines 148-149, it is mentioned that, in the on-bed scenario, the detected object can be “head”. And it can be “body” in the out-of-bed scenario. According to this statement, were different target objects aimed for different scenarios? If so, how the system handle misclassification shall be discussed.
Most importantly, the experiments in the paper only evaluated the effectiveness of the behavior recognition module where the input video is already segmented. The effectiveness of the system where it is used at the target task shall also be investigated. That is the system performance where the input is the whole video sequence of users lying on the bed with such bed-exit activities shall be evaluated.
Reviewer 2 Report
This paper proposed an image-based behavior recognition system to detect three bed related behavior in the ward. It uses RGB video camera to capture NFW image frames in bright light conditions. Yolov4-tiny has been adopted as the algorithm for object detection and tracking. The topic is interesting and the use case is sound. My major concerns are:
1. The light condition may change when the patient needs to sleep, how to detect bed behavior in dark environment? The introduction part listed multiple non-RGB sensors including IR thermal images and wearable sensors to detect movement. I highly suggest the authors to include object detection and tracking algorithms using multiple sensors for bed behavior classification in dark conditions (sensor fusion). If not applicable, at least include the related references [1][2].
2. There is no adjustment or improvement of Yolov4-tiny for the specific use case. At least there could be some other SOTA algorithms for performance comparison. It is suggested to add reference [3] for object detection, and [4] for object tracking.
[1] Correlative Mean-Field Filter for Sequential and Spatial Data Processing, in the Proceedings of IEEE International Conference on Computer as a Tool (EUROCON), Ohrid, Macedonia, July 2017
[2] Distributed mean-field-type filters for Big Data assimilation, in the second IEEE International Conference on Data Science and Systems (HPCC-SmartCity-DSS), Sydney, Australia, Dec, 2016, pp. 1446-1453
[3] Exploring Plain Vision Transformer Backbones for Object Detection
[4] Spiking Transformers for Event-Based Single Object Tracking
Reviewer 3 Report
1. The Abstract must be relevant for paper including the motivation.
2 The scope of the work must be clear also in Abstract. So, what is the scope, the presence or absence in the bed or the movement of patient?
3. The research in this area is very low, only very few papers deal with this known subject. Also, the area is very very narrow, so the interest of readers is not justified.
4. Despite of large introduction (and references), Bed-exit monitoring is practically a sort part and inclusion of large part of object recognition techniques that is a huge area of research is not justified.
6. The references should filter to refer to subject of the paper.
7. The Fig 4-5 is very general and simplest.
8. The "embedded system" is not presented (hardware).
9. The paper devote a large portion of complexity of calculus. What is the justification of this mathematics in a monitoring system?
10 The Eq. 6 is unclear for a reader. There are four situations: off, on, return, nobody. A natural question arise related to image processing and computer vision: there are no much simpler methods to detect presence or absence a one person sin a frame?
The same question is about the sequences presence/absence of one person in a frame.
11. The paper should better organized. There should be clear section to identify how the block works: classifier, controller and decision of behavior.
12. Table 1, and 2 are unclear. The authors should define with example what means TP and TN (These are general statistic definition) in this case, with example. Much more, the classifier should have a class; These classes must be defined in order to know what we classify.
13. The LSTM classifier is hard to be discovered in this large paper, only one sentence or maybe two just before eq. 13. Finally, the reader cannot draw a conclusion about how was used this classifier for an extremely general sentence about LSTM as deep learning method.
14.The original contribution is not clear at all. Especially, the results are not compared with [30] that achieve an accuracy of 99%.
Round 2
Reviewer 2 Report
The revised version has addressed all my concerns, it is ready to publish.
Reviewer 3 Report
All the requirements are fulfilled. The paper can be published.